# Neural effects of continuous theta-burst stimulation in macaque parietal neurons

Maria C Romero[1], Lara Merken[1], Peter Janssen[1]\*[†], Marco Davare[2]\*[†]

[1]Laboratorium voor Neuro- en Psychofysiologie, The Leuven Brain Institute, Leuven, Belgium; [2]Faculty of Life Sciences and Medicine, King's College London, London, United Kingdom

**Abstract** Theta-burst transcranial magnetic stimulation (TBS) has become a standard non-invasive technique to induce offline changes in cortical excitability in human volunteers. Yet, TBS suffers from a high variability across subjects. A better knowledge about how TBS affects neural activity in vivo could uncover its mechanisms of action and ultimately allow its mainstream use in basic science and clinical applications. To address this issue, we applied continuous TBS (cTBS, 300 pulses) in awake behaving rhesus monkeys and quantified its after-effects on neuronal activity. Overall, we observed a pronounced, long-lasting, and highly reproducible reduction in neuronal excitability after cTBS in individual parietal neurons, with some neurons also exhibiting periods of hyperexcitability during the recovery phase. These results provide the first experimental evidence of the effects of cTBS on single neurons in awake behaving monkeys, shedding new light on the reasons underlying cTBS variability.

**\*For correspondence:**
peter.janssen@kuleuven.be (PJ);
marco.davare@kcl.ac.uk (MD)

[†]These authors contributed equally to this work

**Competing interest:** The authors declare that no competing interests exist.

## Editor's evaluation

This paper provides a fundamental advance on our understanding of the effects of a brain stimulation technique, continuous theta burst transcranial magnetic stimulation, that is widely used across a variety of subfields within human neuroscience. With convincing methodological rigor, the authors provide important validation of mechanism of action that produce the long-lasting effects of stimulation while simultaneously providing clues that speak to the variability observed in prior studies.

## Introduction

Repetitive transcranial magnetic stimulation (rTMS) protocols, such as continuous theta-burst stimulation (cTBS), represent a non-invasive way to reduce cortical excitability in human volunteers (*Huang et al., 2005*) and to explore neuroplasticity in several patient populations (*Edwards et al., 2006*; *Oberman et al., 2010*; *Orth et al., 2010*; *Suppa et al., 2011*; *Koch et al., 2012*; *Munneke et al., 2013*; *Opie et al., 2013*; *Chuang et al., 2014*; *Mori et al., 2014*; *Suppa et al., 2014a*; *Suppa et al., 2014b*). However, a number of major hurdles prevent its widespread application in basic research and in clinical care. For example, despite more than 10 years of studies in human volunteers, it is still unclear how neural activity changes in the cortex after cTBS (*Pitcher et al., 2021*). It is generally assumed that the reduction in neuronal excitability may resemble the changes observed in long-term depression (LTD; *Fitzgerald et al., 2006*; *Thut and Pascual-Leone, 2010*; *Vlachos et al., 2012*), but no study has ever tested in vivo whether this is true. The reduction in the amplitude of the motor evoked potential (MEP) after cTBS also grows over time (*Huang et al., 2005*), which does not resemble the immediate reduction in neuronal excitability after conditioning observed in LTD. Moreover and more importantly, the physiological effects of cTBS are notoriously variable, with inhibitory effects in some subjects and facilitatory effects in other subjects (*Hamada et al., 2013*). Several factors could contribute to this

variability (reviewed in *Suppa et al., 2016*). Genetic variation, differences in the intracortical network activated by the TMS pulses (measured in the so-called late I-waves), previous levels of activity, circadian effects, and differences in the brain state may all contribute to the inter- and intra-individual variability reported in cTBS studies.

We sought answers to the aforementioned questions by recording single-neuron activity before and after cTBS in parietal cortex of awake behaving rhesus monkeys. We probed the neuronal excitability using single-pulse TMS (sTMS), and measured how this excitability changed up to 2 hr after cTBS. To the extent possible, we standardized the factors potentially contributing to the known variability associated with cTBS: the positioning of the coil on the skull, the level of ongoing motor activity, the stimulation hour, and the brain state were equalized as much as possible across sessions and animals. We observed highly reproducible changes in neuronal excitability, in which individual neurons would progress through phases of hypo- and hyperexcitability, in some cases followed by recovery. At the population level, the reduction in neuronal excitability grew over time, reaching its maximum 30–40 min after cTBS, consistent with studies in humans. Thus, in a standardized experimental setting, cTBS induced reproducible and long-lasting changes in neuronal excitability.

## Results

We recorded single-unit activity in parietal area PFG of two monkeys during sTMS in 90 experimental sessions. Across these sessions, a total of 86 neurons (51 neurons in monkey Y; 35 in monkey A) showed a significant increase in their firing rate in response to sTMS, and were further recorded before and after cTBS during passive fixation.

### Effect of cTBS on individual neurons

Before applying cTBS, we tested the excitability of each neuron using sTMS administered at light onset above the object (*Figure 1*).

*Figure 2* illustrates, with two example neurons, the typical results obtained in this study. The first example neuron (*Figure 2A*) generated a brief burst of action potentials almost immediately after sTMS (top row) but did not respond to light onset in the absence of sTMS. After 20 s of cTBS, however, the excitability of this neuron was markedly reduced (second row, sTMS-evoked response pre- compared to post-cTBS: z=3.88, p=1.06 e−04, r=0.60; Wilcoxon test). Indeed, in the first 10 min post- CTBS, the activity in high-stimulation trials did not differ anymore from the activity in no-stimulation trials (z=1.81, p=0.07, r=0.17). Over the subsequent time intervals, the excitability of this neuron gradually recovered (no-stimulation compared to stimulation trials; 20 min post-cTBS: z=4.99, p=6.01e−07, r=0.46; 30 min post-cTBS: z=6.58, p=4.57e−11, r=0.60; 40 min post-cTBS: z=5.27, p=1.35e−07, r=0.48; 50 min post-cTBS: z=4.74, p=2.17e−06, r=0.43), but only after 60 min did the excitability of the neuron return to the pre-cTBS level (sTMS-evoked response pre- compared to 60 min post-cTBS: z=1.63, p=0.103, r=0.26). Thus, cTBS caused a marked and immediate reduction in excitability in this parietal example neuron, which recovered over the course of 1 hr.

The second example neuron also responded to sTMS before we applied cTBS (*Figure 2B*, top panel, notice also the weak response to light onset in the no-stimulation trials), but was not affected by cTBS in the first 10 min after cTBS (*Figure 2B*, second panel, sTMS-evoked response pre- compared to post-cTBS: z=−0.61, p=0.548, r=−0.10). Only in the 20 and 30 min intervals after cTBS, this neuron's excitability dropped considerably (no significant difference between stimulation and no-stimulation trials at 20 min post-cTBS: z=−0.84, p=0.399, r=−0.07, and at 30 min post-cTBS: z=−0.20, p=0.839, r=−0.02), although its visual response (to light onset: baseline vs. post-light onset activity in no-stimulation trials; z=−4.72, p=2.32e−06, r=−0.32) remained. Surprisingly, in the 40 and 50 min intervals, the excitability of this neuron temporarily increased significantly (sTMS-evoked response pre-compared to post-cTBS at 40 min post-cTBS: z=−2.24, p=0.025, r=−0.355 and at 50 min post-cTBS: z=−2.68, p=0.007, r=−0.42). Only in the 60 min interval did the neuron return to its pre-cTBS excitability level (z=−2.61, p=0.08, r=−0.41, pre-compared to 60 min post-cTBS in the first 40 ms after TMS onset). A third example neuron (*Figure 3A*) showed yet another pattern of excitability changes after cTBS. The strong burst of activity evoked by sTMS was immediately reduced after cTBS (*Figure 3A*, compare top panel with second panel: z=3.64, p=2.741e−04, r=0.68). However, this reduction in excitability grew markedly over time and reached its peak only after 50 min. At the end of our standard 60 min

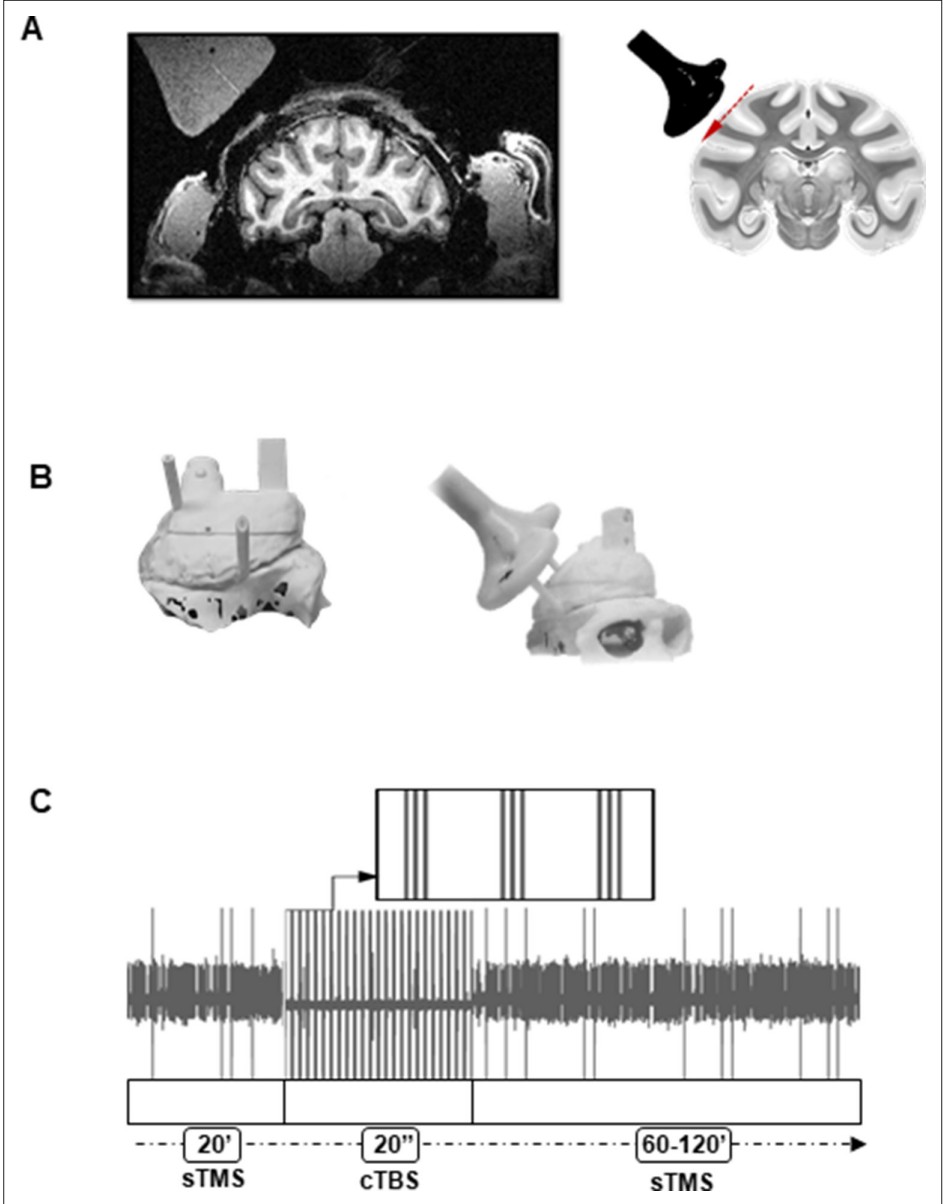

**Figure 1.** Brain targeting and stimulation protocol. (**A**) Left: Anatomical magnetic resonance image performed with a dummy coil, reproducing the position and angle of the Magstim coil during recordings (45°ee angle with respect to the vertical). Right: Coronal view of the monkey brain indicating the location of the TMS coil with respect to parietal area PFG. The red, dashed line indicates the trajectory of the electrode during recordings. For every experiment, a D25 mm figure-of-eight TMS coil (in black) was rigidly anchored to the monkey's implant. (**B**) Three-dimensional models of the monkey's skull and implant (left: top view, right: lateral view). Prior to the experiment, two guiding rods were attached to the monkey's head implant based on MRI estimations of the cortical target coordinates. This allowed a precise and reproducible coil positioning across experimental sessions. (**C**) Example of the raw signal recorded during a typical stimulation session. The high voltage, saturated peaks indicate the stimulation time stamps. TMS was administered in three different epochs corresponding, in this order, to: single-pulse TMS (sTMS) applied at light onset (20 min), cTBS (20 s), and again, sTMS (60–120 min). cTBS, continuous theta-burst stimulation; MRI, magnetic resonance imaging; TMS, transcranial magnetic stimulation.

recording epoch, the neuron did not respond anymore to sTMS, without any sign of recovery, although we could still sporadically detect spikes (*Figure 3B*). In addition to the growing suppression of the sTMS response in the first hour after cTBS, we also observed a significant reduction in the baseline activity in this neuron (by 91%, pre-compared to 40 min post-cTBS: z=3.64, p=1.251e–05, r=0.65). Because the activity of the neuron was extremely reduced, we verified that the neuron was not lost

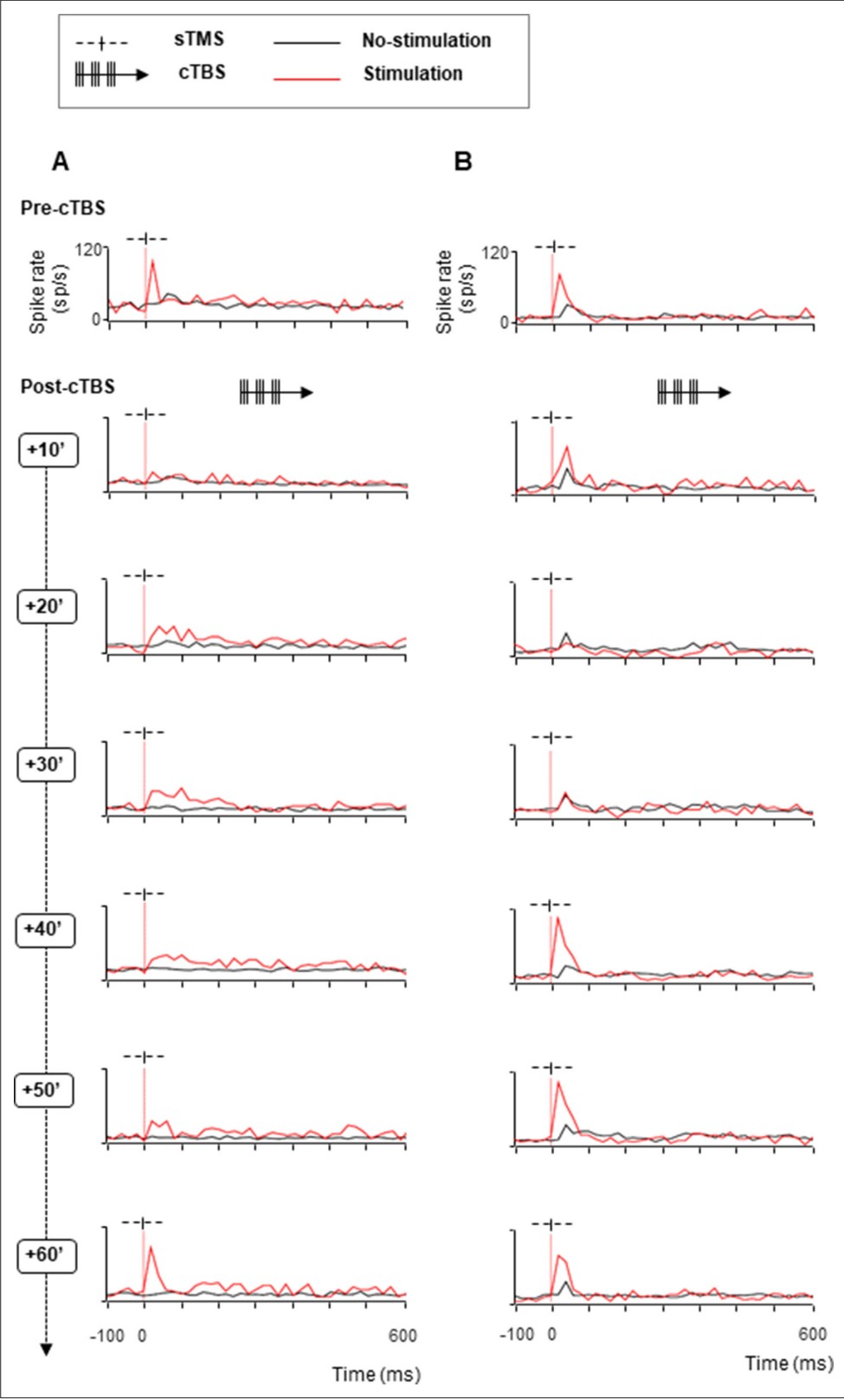

**Figure 2.** Effect of cTBS on neuronal excitability: example neurons with recovery. We tested the excitability of each neuron using single-pulse TMS (sTMS) administered before and after cTBS. Stimulation (red line plots) and no-stimulation trials (black line plots) were randomly interleaved during the passive fixation task. The red, dotted line indicates the sTMS onset (aligned to light onset). (**A**). Spike rate of an example neuron exhibiting a short-

*Figure 2 continued on next page*

*Figure 2 continued*

lasting, excitatory response to sTMS (top row). cTBS caused a marked and immediate reduction in excitability (row 2), which disappeared over the course of 1 hr (rows 3–7). (**B**). Second example neuron. As in (**A**), this neuron responded to sTMS before cTBS. However, there was no effect of cTBS until 20 min post-stimulation (row 3). During the recovery phase, the neuron showed a period of hyperexcitability (40–50 min post-cTBS; rows 5 and 6). cTBS, continuous theta-burst stimulation; sTMS, single-pulse transcranial magnetic stimulation.

during the recording session by comparing the spike waveforms recorded before (*Figure 3B*, upper panel) and 60 min after cTBS (*Figure 3B*, lower panel). The firing rate of this neuron in the 60 min post-cTBS interval was reduced to barely 4 spikes/s, but the spike waveform was virtually identical compared to the pre-cTBS epoch, which confirms that we still recorded from the same neuron. Notice also a very weak response to sTMS at 60 min post-cTBS. Overall, cTBS induced a compelling reduction in the excitability of parietal neurons, with highly variable onset time and recovery.

## Population analysis

Because the results were highly comparable in the two animals, we pooled all neurons recorded, but we also provide statistics for each animal separately (see *Figure 4—figure supplement 1*). In the first 10 min post-cTBS, almost half of the PFG neurons (43%; 39% in monkey Y and 49% in monkey A) showed a significant (two-sided Wilcoxon rank sum test p<0.05) change in their sTMS response (either hypo- or hyperexcitability). However, the proportion of neurons in which cTBS affected the sTMS response gradually increased over time, such that in the 60 min post-cTBS interval, virtually all neurons (85/86) were significantly affected by cTBS (*Table 1*; with Bonferroni correction, the proportions of affected neurons grew from 24% at 10 min post-cTBS to 62% at 60 min post-cTBS). Similarly, cTBS induced an immediate and significant change in the baseline activity (i.e., the spike rate before sTMS) in about one third of the neurons (34%, two-sided Wilcoxon rank sum test p<0.05), which reached its maximum 1 hr post-cTBS (93% of the neurons, *Table 1*). Overall, 21% of the neurons showed an effect both in the sTMS response and the baseline activity in the first 10 min post-cTBS, compared to 83% of the neurons showing a combined effect 1 hr post-cTBS.

On average, cTBS significantly reduced the neuronal excitability assessed with sTMS, an effect that grew over the standard 1 hr post-cTBS period (*Figure 4* and *Figure 4—figure supplement 1*). In the first 10 min interval post-cTBS, the average sTMS response was 24% lower than pre-cTBS (z=2.11, p=0.03, r=0.16), and this reduction peaked and plateaued 30–50 min post-cTBS (32% reduction). Similarly, the reduction in the average baseline activity induced by cTBS emerged already in the first 20 min post-cTBS (by 14%; z=1.42, p=0.04, r=0.15), became more apparent 40 min post-cTBS (28% reduction, z=3.23, p=0.001, r=0.25), and recovered partially at 60 min post-cTBS (20% reduction, two-sided Wilcoxon rank sum tests comparing the pre-cTBS cell response with that measured at 40 min post-cTBS; for the two animals at 40 min: z=2.66, p=0.01, r=0.26 for monkey Y and z=1.86, p=0.06, r=0.22 for monkey A).

The cTBS effect was highly similar in both animals (maximum 32% reduction in sTMS response in monkey Y, and 29% reduction in sTMS response in monkey A), but the timing of the effect differed slightly, since the maximum reduction appeared at 30 min post-cTBS in monkey Y and at 40 min post-cTBS in monkey A (*Figure 4—figure supplement 1*).

To investigate whether the cTBS effect on the neuronal sTMS response and the cTBS effect on the baseline activity were based on similar neuronal mechanisms, we plotted the changes in the sTMS response and in the baseline activity against each other and calculated the correlation between the two effects in every epoch post-cTBS (*Figure 5*; see *Table 2*). In each individual animal (data not shown) and across all neurons combined, we measured significant and high correlation coefficients ranging from 0.54 (at 60 min post-cTBS) to 0.80 (at 30 min post-cTBS), suggesting that cTBS induced a general change in neuronal excitability, which was manifested in both sTMS response changes and alterations in spontaneous activity. This correlation increased significantly over time (20, 30, 40, and 50 min interval compared to the first 10 min post-cTBS), indicating that the cTBS effect became more apparent in both the baseline activity and the sTMS evoked responses. At 60 min post-cTBS, the correlation declined significantly compared to the preceding intervals, most likely because neurons started to recover from the reduction in excitability induced by cTBS.

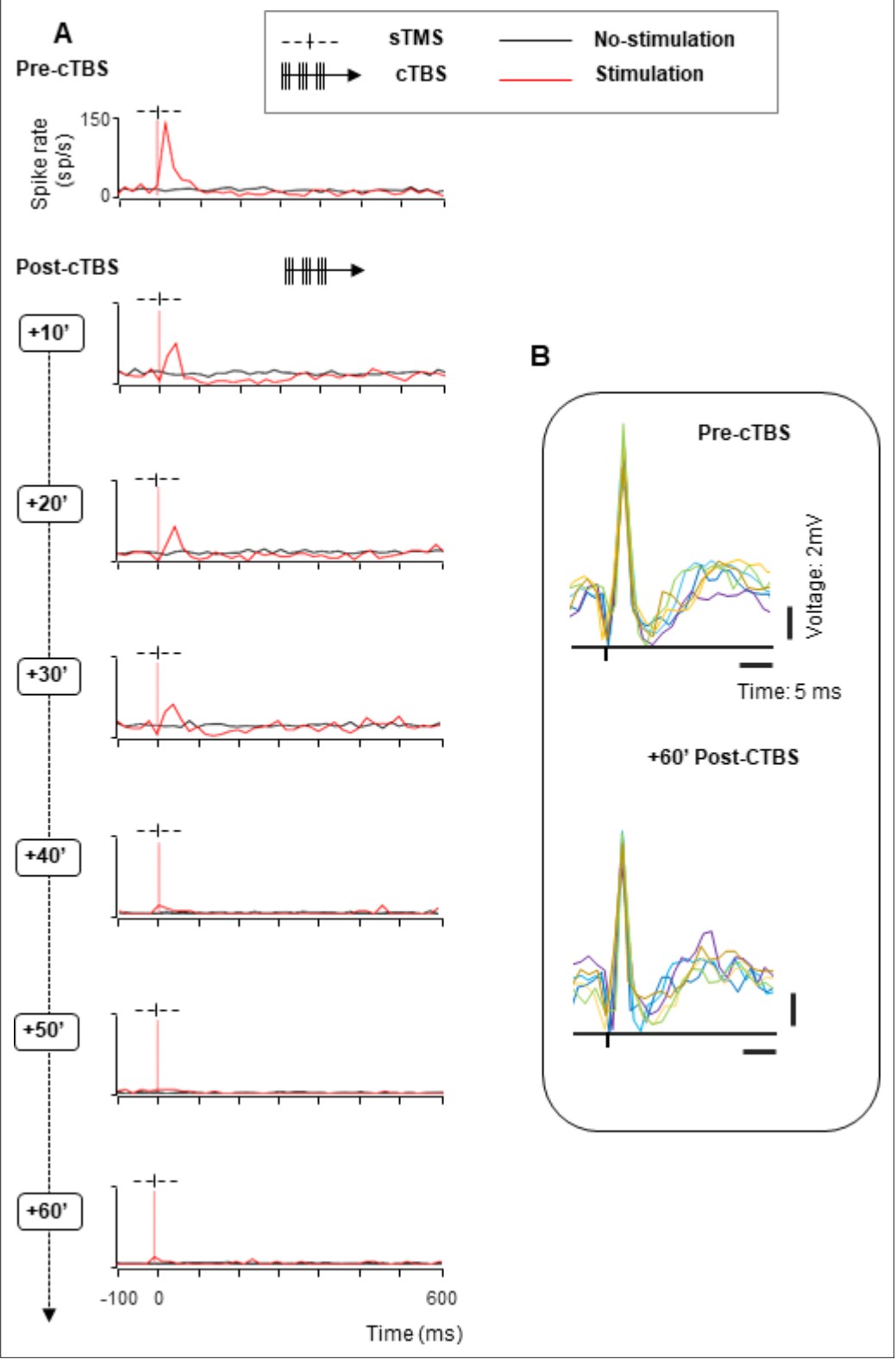

**Figure 3.** Effect of cTBS on neuronal excitability: example neuron without recovery. (**A**) Spike rate of an example neuron with immediate reduced excitability after cTBS (rows 2–4). A stronger reduction, accompanied by a significant decrease of the baseline activity appeared later (40 min post-cTBS; row 4), continuing until the end of the session (60 min post-cTBS; row 7). Same conventions as in *Figure 2*. (**B**) Waveforms of the example neuron. Voltage graph showing the overlapped spike waveforms extracted from six consecutive trials (represented in

*Figure 3 continued on next page*

different colors), recorded at two different time intervals (upper panel: pre-cTBS; lower panel: 60 min post-cTBS). cTBS, continuous theta-burst stimulation.

*Figure 5* also illustrates that, although cTBS in general reduced neuronal excitability (53% of the neurons showed hypoexcitation in all time epochs post-cTBS), transient phases of hyperexcitability were not uncommon in our neuronal population. Overall, almost half of the neurons (47%) were hyperexcitable in at least one interval post-cTBS (*Table 1*). The large majority of these neurons (91%) were initially less excitable followed by an epoch of hyperexcitability, as the example neuron in *Figure 2B*. However, in a small number of neurons, hyperexcitability appeared immediately after cTBS, either as the only effect throughout the entire recording session (6%) or followed by hypoexcitability after a variable time interval (10–40 min, 3%). Significant increases in baseline activity were also not uncommon, since 44% of the neurons showed hyperexcitability in at least one epoch post-cTBS (*Table 1*).

cTBS may not only influence the average neuronal response to sTMS, but also the spike timing, that is, the variability of the interspike intervals. We could not detect any significant effect of cTBS on the variance to mean ratio (the Fano factor) of the neurons, nor on the distribution of the interspike intervals (ISIs, data not shown). However, the power spectrum of the spike trains changed significantly after cTBS, both in the stimulation and in no-stimulation trials (*Figure 6*). In every epoch post-cTBS, the power in the low frequencies (below 5 Hz) was significantly reduced compared to pre-cTBS. Thus, cTBS also induced changes in the low-frequency oscillatory activity of parietal neurons.

## Effect of cTBS on task-related neural activity

Since a motor task may cause small movements of the electrode and since we wanted to give priority to the stability of the neural recordings, we chose to use a passive fixation task for the recordings, in which the monkeys were required to simply fixate an object illuminated in front of them to obtain a fluid reward. Despite the absence of a grasping movement, a subset of the neurons we recorded in PFG showed significant task-related activity (i.e., object responses) after light onset (N=18). The presence of object responses allowed us to test the effect of cTBS on neuronal excitability without applying sTMS. To capture all task-related responses, we tested the effect of cTBS in two intervals, an early (0–80 ms after light onset) and a late interval (100–500 ms after light onset) and analyzed the response difference in the 40 ms around the maximal response. The later task-related activity (100–500 ms after light onset) differed significantly compared to the pre-cTBS epoch in every epoch from 30 to 60 min post-cTBS, whereas the early interval after light onset did not show a significant

**Table 1.** Proportions of neurons with a cTBS effect in different time epochs.

|  | 10 min post-cTBS | 20 min post-cTBS | 40 min post-cTBS | 60 min post-cTBS |
|---|---|---|---|---|
| sTMS effect | 43% (37/86)<br>*24% (21/86)<br>Y: 20/51; A: 17/35<br>*Y: 9/51; A: 12/35 | 53% (46/86)<br>*33% (28/86)<br>Y: 23/51; A: 23/35<br>*Y: 14/51; A: 14/35 | 86% (74/86)<br>*57% (49/86)<br>Y: 43/51; A: 31/35<br>*Y: 26/51; A: 23/35 | 99% (85/86)<br>*62% (53/86)<br>Y: 50/51; A: 35/35<br>*Y: 29/51; A: 24/35 |
| Baseline effect | 34% (29/86)<br>*22% (19/86)<br>Y: 17/51; A: 12/35<br>*Y: 12/51; A: 7/35 | 50% (43/86)<br>*33% (28/86)<br>Y: 24/51; A: 19/35<br>*Y: 18/51; A: 10/35 | 77% (66/86)<br>*44% (38/86)<br>Y: 40/51; A: 26/35<br>*Y: 24/51; A: 14/35 | 93% (80/86)<br>*50% (43/86)<br>Y: 48/51; A: 32/35<br>*Y: 30/51; A: 13/35 |
| Hyperexcitability in sTMS response | 10% (9/86)<br>*3% (3/86)<br>Y: 6/51; A: 3/35<br>*Y: 2/51; A: 1/35 | 10% (9/86)<br>*7% (6/86)<br>Y: 6/51; A: 3/35<br>*Y: 4/51; A: 2/35 | 19% (16/86)<br>*15% (13/86)<br>Y: 10/51; A: 6/35<br>*Y: 8/51; A: 5/35 | 28% (24/86)<br>*23% (20/86)<br>Y: 15/51; A: 9/35<br>*Y: 12/51; A: 8/35 |
| Hyperexcitability in baseline activity | *10% (9/86)<br>6% (5/86)<br>Y: 5/51; A: 4/35<br>*Y: 3/51; A: 2/35 | *12% (10/86)<br>10% (9/86)<br>Y: 5/51; A: 5/35<br>*Y: 5/51; A: 4/35 | *15% (13/86)<br>13% (11/86)<br>Y: 8/51; A: 5/35<br>*Y: 7/51; A: 4/35 | *24% (21/86)<br>21% (18/86)<br>Y: 13/51; A: 8/35<br>*Y: 11/51; A: 7/35 |

Results on the Wilcoxon test without Bonferroni correction calculated for 10′ intervals.
*Results on the Wilcoxon test with Bonferroni correction calculated for 10′ intervals.

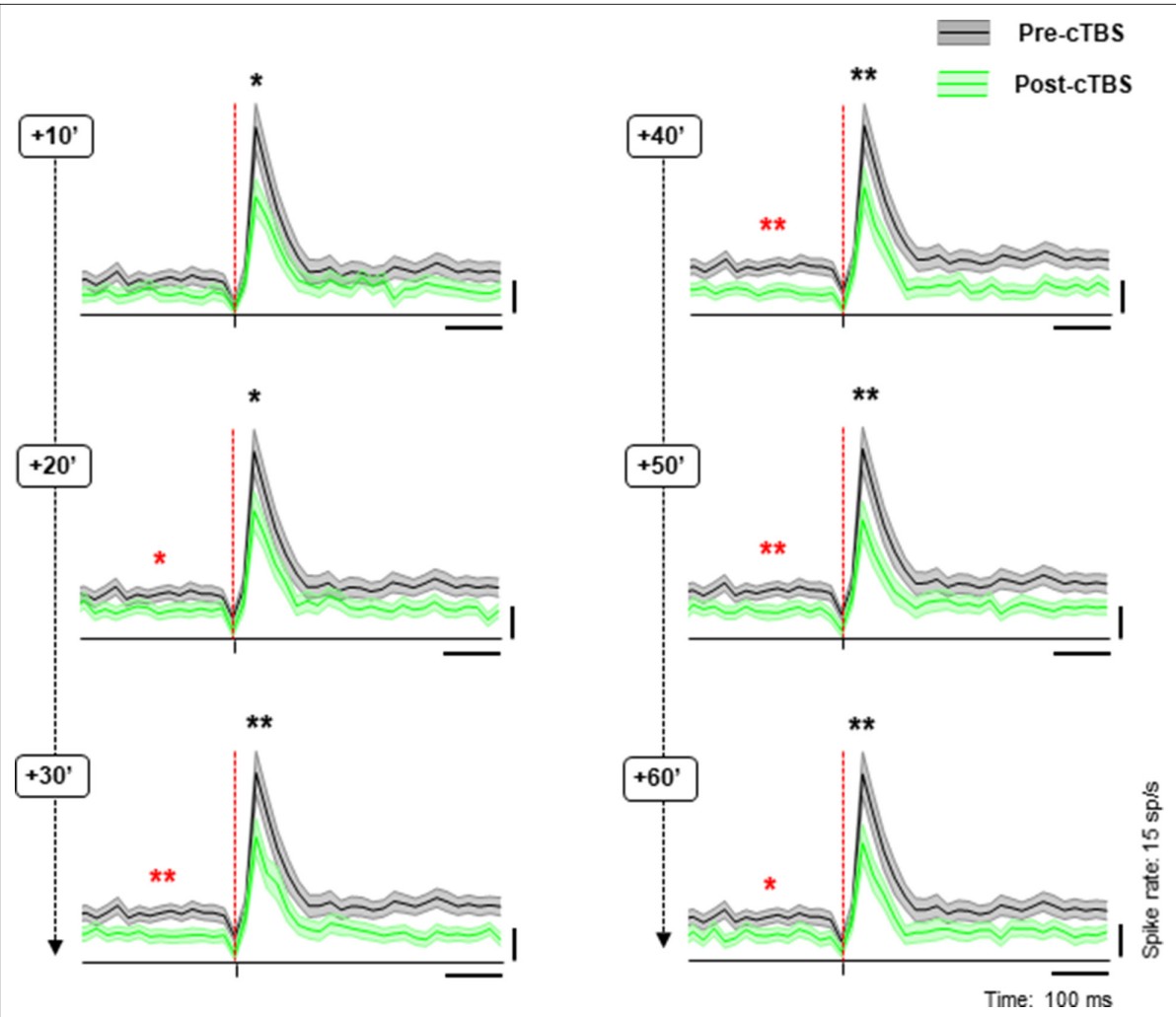

**Figure 4.** Population response to cTBS. Average sTMS responses for all neurons at different time periods post-cTBS (green) compared to pre-cTBS (black), when sTMS was applied at light onset during passive fixation. Shading displays ±the standard error across neurons (N=86). The red, dashed line indicates the sTMS pulse, aligned to light onset. The asterisks specify the statistical significance (two-sided Wilcoxon rank sum test; *p≤0.05; **p≤0.01) for changes in both the baseline activity (red) and the sTMS-evoked response (black). cTBS, continuous theta-burst stimulation; sTMS, single-pulse transcranial magnetic stimulation.

The online version of this article includes the following figure supplement(s) for figure 4:

**Figure supplement 1.** Population response to cTBS for monkeys Y and A, separately.

difference in any epoch post-cTBS (Wilcoxon signed rank test, *Figure 7A*). When probed with sTMS, this subpopulation of task-related neurons behaved similar to the rest of the population, since the neuronal excitability was significantly reduced at 30 min post-cTBS (*Figure 7B*). Overall, cTBS also induced a reduction in task-related activity in parietal neurons, comparable to the effects observed with sTMS.

## Effect of cTBS on neuronal excitability beyond 1 hr

Our standard recording time was 1 hr after cTBS (plus approximately 20 min of recordings pre-cTBS). However, in a subpopulation of neurons (N=34; *Figure 8A*), we could test the effect of cTBS on neuronal excitability for up to 90 min post-cTBS. Because neurons can be lost in the course of such a long time interval due to small brain movements, we only included units which the signal-to-noise ratio was at least 5:1 for the entire duration of the recording session, and we compared the spike waveforms pre- and post-cTBS to verify that the neuron was still present (see example spike waveforms in *Figure 8B*). Even 90 min after cTBS, most neurons (18/34, 53%) showed significantly

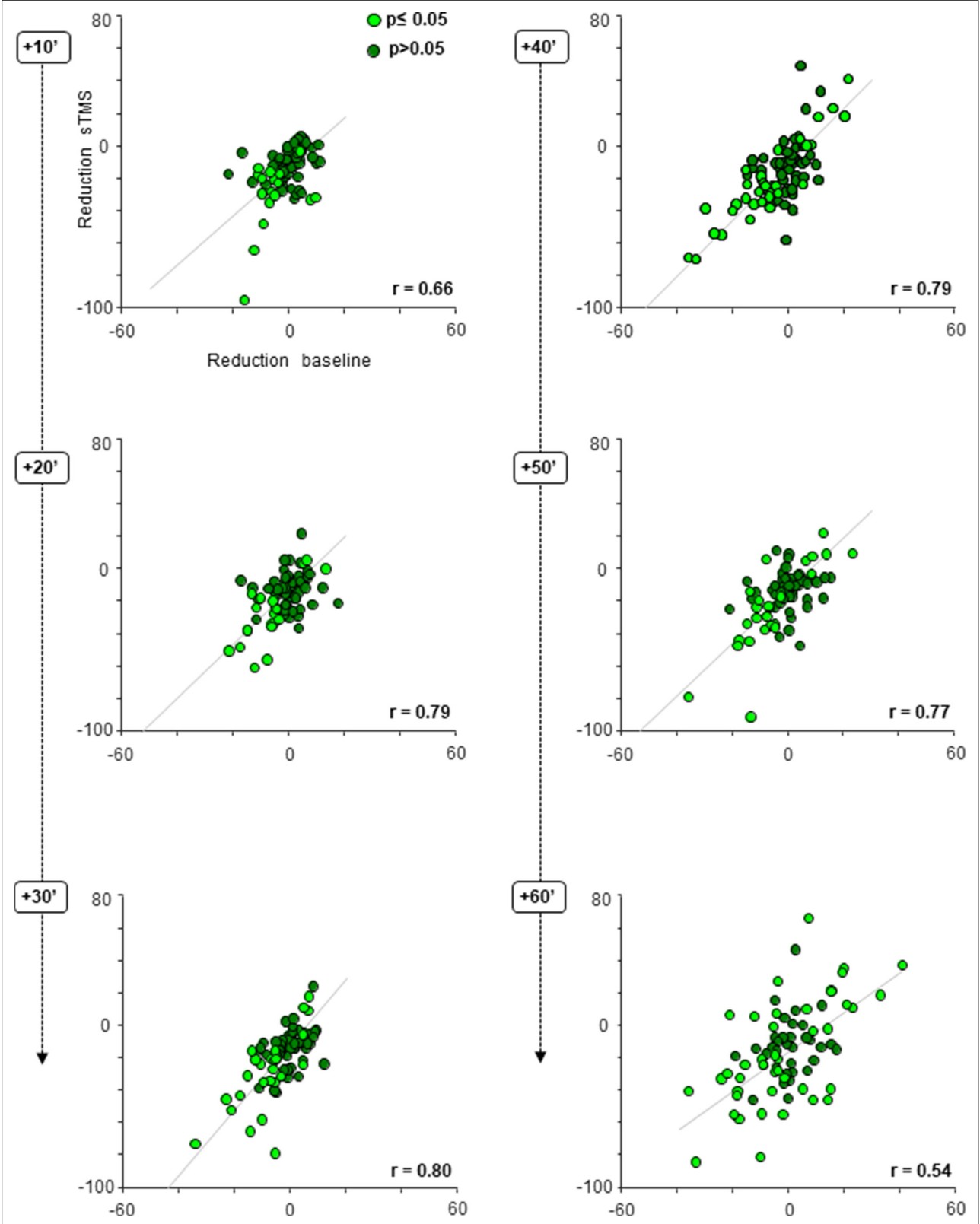

**Figure 5.** cTBS effect on neuronal excitability; scatter plots. Average response difference between the pre- and post-cTBS period plotted against the difference in the baseline activity pre- versus post-cTBS, at six consecutive, 10 min intervals post-cTBS. Each gray line indicates the least squares line (best fit). For every panel, each colored dot represents a PFG neuron. Light green dots reflect neurons showing a statistically significant (two-sided Wilcoxon rank sum test p≤0.05) change in both their sTMS and baseline response (either hypo- or hyperexcitability) post-cTBS. Dark green dots indicate

*Figure 5 continued on next page*

*Figure 5 continued*

neurons without statistically significant effect (two-sided Wilcoxon rank sum test p>0.05). cTBS, continuous theta-burst stimulation; sTMS, single-pulse transcranial magnetic stimulation.

reduced sTMS responses compared to pre-cTBS, and the average normalized sTMS-evoked response was significantly reduced at this late time period post-cTBS (z=1.99, p=0.02, r=0.24). Moreover, we also detected a significant reduction in neuronal activity after the sTMS-evoked burst (in the interval 200–500 ms after sTMS; z=−2.58, p=0.01, r=−0.32). A smaller subset of neurons (N=15) was monitored for up to 2 hr post-cTBS, but again no recovery was detectable in this subpopulation (data not shown). Thus, the cTBS-induced reduction in neuronal excitability is very prolonged and may not recover for several hours post-cTBS.

## Discussion

We observed a pronounced and long-lasting reduction in neuronal excitability after cTBS in macaque parietal neurons. This reduced excitability of individual neurons grew over time, consistent with previous studies in humans, but some neurons exhibited periods of hyperexcitability during the recovery phase. These results provide the first experimental evidence on the neural effects of cTBS on single neurons in awake behaving monkeys. Importantly, while we did not control for remote or network cTBS effects as per scope of this study, our findings that cTBS did not systematically lead to inhibition of neuronal firing rate, that hypo- and hyperexcitability periods occurred at different time points in different neurons and that neuronal activity recovered to baseline, pre-cTBS levels, on average an hour post-cTBS, allow us to argue against nonspecific effect of the cTBS intervention. Moreover, in our previous work investigating the behavioral effects of cTBS on monkeys (*Merken et al., 2021*), we showed the differential effect of effective versus sham cTBS on motor performance, providing indirect evidence about the specificity of the underlying neural cTBS effects found in this study.

Our basic findings, both at the single-cell level and at the behavioral level (see *Merken et al., 2021*), are remarkably consistent with the known effects of cTBS over primary motor cortex in human volunteers (*Huang et al., 2005*). The average amplitude of the MEP—a measure of neuronal excitability in the primary motor cortex—is unaltered in the first minutes after cTBS, then gradually declines and recovers 25 (after 20 s cTBS) or 60 min (after 40 s cTBS) later. In parallel, the reaction time significantly increases 10 min after cTBS over primary motor cortex (*Huang et al., 2005*). The time course of the neural effects we measured after 20 s of cTBS in monkeys seems to be more similar to the 40 s cTBS protocol in humans, possibly due to the thinner skull of monkeys. Nevertheless, it should also be noted that we applied cTBS over parietal cortex, whereas most cTBS studies in humans have targeted primary motor cortex (but see *Davare et al., 2010*). Overall, the consistency of our findings with the human literature strongly suggests that single-cell recordings after cTBS in awake macaque monkeys represent a valid approach to understanding the neural effects of cTBS.

The crucial advantage of our approach is that we can measure the wide range of changes in excitability in individual neurons, while the MEP amplitude represents an excitability measure of a large population of neurons. In addition, we could demonstrate that the inhibitory influence of cTBS progressively recruits more neurons over time, even neurons that were initially unaffected. Finally, we showed that cTBS induces a general reduction in neuronal excitability, which appears in changes in baseline firing rate, sTMS-evoked responses, low-frequency oscillatory activity, and task-related activity. It is noteworthy that the neuronal object responses we measured here in PFG were relatively slower and longer-lasting, compared to neighboring area AIP (e.g., *Pani et al., 2014*).

Extracellular recordings do not easily allow determining the cell type of the unit that is being recorded based on the spike waveform (see, e.g., *Woloszyn and Sheinberg, 2012* compared to *Vigneswaran et al., 2011*), nor the cortical layer in which these units were recorded. In general, we

**Table 2.** Correlations between the sTMS-evoked and the baseline effect with confidence intervals in different time epochs.

| 95% CI | +10′ | +20′ | +30′ | +40′ | +50′ | +60′ |
|---|---|---|---|---|---|---|
| Lower bound | 0.51 | 0.69 | 0.70 | 0.69 | 0.67 | 0.36 |
| r | | 0.66 | 0.79 | 0.80 | 0.79 | 0.77 | 0.54 |
| Upper bound | 0.76 | 0.86 | 0.86 | 0.86 | 0.85 | 0.68 |

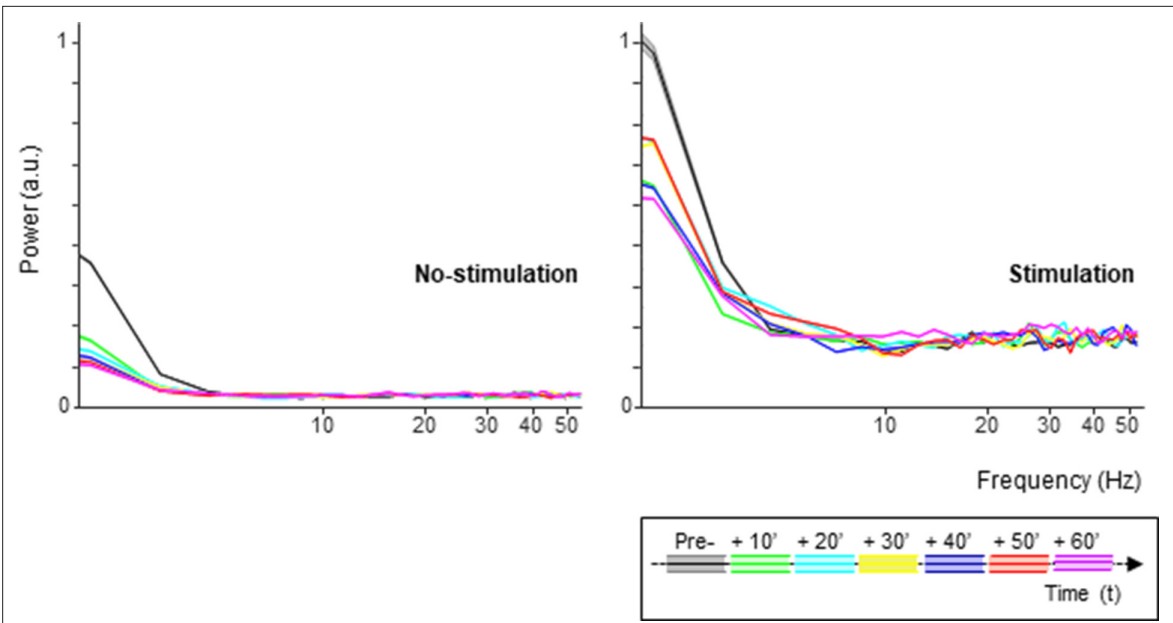

**Figure 6.** Spike oscillations analysis. Spectral power of the single-unit activity in both the no-stimulation (left panel) and stimulation (sTMS) trials (right panel), divided in 10 min intervals pre- and post-cTBS. Each color line indicates a different time interval. The shading in the graph represents ±the standard error. cTBS, continuous theta-burst stimulation; sTMS, single-pulse transcranial magnetic stimulation.

recorded from neurons located immediately under the TMS coil and generated large spike wave-forms, that we could monitor for more than 1 hr and up to 2 hr post-cTBS. However, the large range of effects we observed in our population of neurons may suggest that cTBS exerts specific effects on different cell types, on neurons in different layers, and in a different orientation and/or location with respect to the TMS coil. In nonhuman primates, future studies could address these questions with advanced techniques such as calcium imaging (*Ikezoe et al., 2013*; *Tang et al., 2020*).

Previous studies have suggested that cTBS may induce LTD-like effects on cortical synapses (*Huang et al., 2011*). LTD, a widespread phenomenon driving synaptic plasticity both in subcortical structures and in the cortex, is typically induced by low-frequency stimulation (LFS; at 1 Hz), and its underlying molecular mechanisms may be very diverse depending on brain area and developmental stage (*Massey and Bashir, 2007*). In visual cortex, 15 min of 1 Hz stimulation induces LTD of synaptic responses (*Kirkwood and Bear, 1994*). However, the LTD effect studied in cortical slices appears immediately after the end of LFS and remains constant for up to half an hour. In contrast, the reduction in neuronal excitability we observed after cTBS grew gradually and reached its maximum 30–50 min after cTBS, similar to the reduction in the amplitude of the MEP observed after cTBS in humans (*Huang et al., 2005*). Almost half of the neurons in our sample were not affected at all in the first epoch post-cTBS, but became less excitable in the hour following cTBS. Therefore, the time course of the cTBS effect we measured does not seem to be compatible with a pure LTD effect. Instead, the gradual reduction in neuronal excitability seems more consistent with an increase in the local concentration of the inhibitory neurotransmitter GABA, which may slowly spread in the cortical area under the TMS coil. It is noteworthy that magnetic resonance spectroscopy has demonstrated an increase in GABA in human motor cortex after cTBS (*Stagg et al., 2009*). In the nonhuman primate model, future studies will be able to investigate in more detail which molecular mechanisms are responsible for the effect of cTBS.

Unexpectedly, some neurons also showed a period of hyperexcitability after an initial phase of reduced excitability caused by cTBS. We interpret these findings as evidence that nearby inhibitory interneurons showed a different time course of recovery after cTBS, such that at some point post-cTBS the neuronal excitability had recovered while the normal inhibitory inputs to the neuron were still less active. In a previous study (*Romero et al., 2019*), we also observed that TMS affects both large pyramidal neurons and small (inhibitory) interneurons. The temporary hyperexcitability in our data may also be related to the rare occurrence of seizures after TMS (*Lerner et al., 2019*).

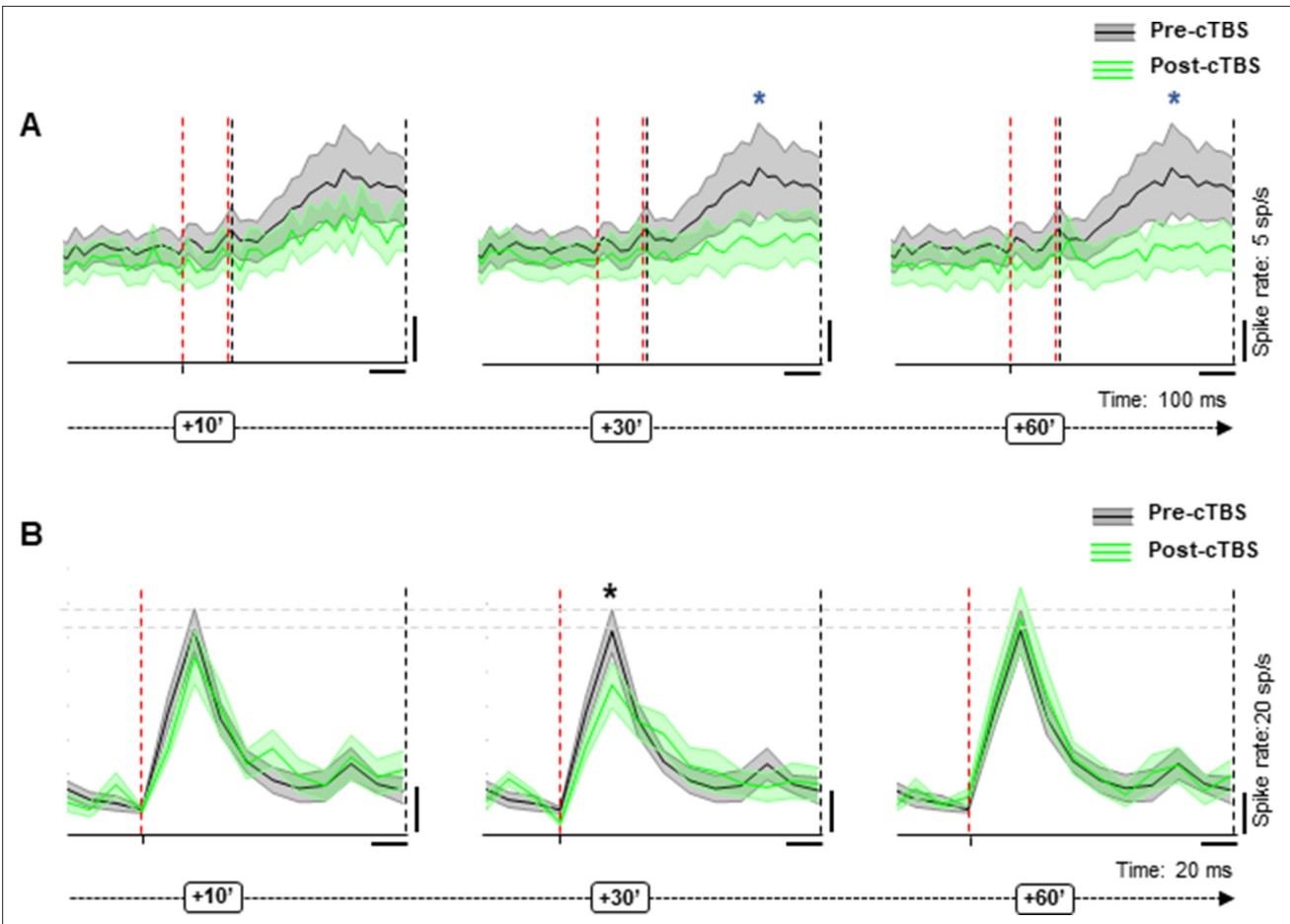

**Figure 7.** Effect of cTBS on task-related activity. (**A**) Pre (black)- versus post-cTBS (green) activity in no-stimulation (sTMS) trials at two different response intervals (30 and 60 min post-cTBS). Shading displays ±the standard error across neurons (N=18). The red, dashed line indicates the onset and offset of the first interval analyzed (early interval: 20–100 ms after light onset); the black, dashed line delimits the second interval of analysis (late: 120–500 ms after light onset). The blue asterisks specify the statistical significance (two-sided Wilcoxon signed rank test; *p≤0.05). (**B**) Response of the same neuronal subpopulation during sTMS trials. Shading displays ±the standard error (N=18). The red, dashed line indicates the sTMS pulse, aligned to light onset. The asterisks specify the statistical significance (two-sided Wilcoxon rank sum test; *p≤0.05). cTBS, continuous theta-burst stimulation; sTMS, single-pulse transcranial magnetic stimulation.

Because we concentrated on measuring the changes in neuronal excitability after cTBS, we did not map the area of cortex under the coil that was affected by cTBS. sTMS affects a surprisingly small volume of cortex, which we estimated to be not larger than 2 by 2 by 2 mm (***Romero et al., 2019***). If our hypothesis of the spreading of GABA is correct, we expect that a slightly larger volume of cortex will be affected by cTBS. Furthermore, our previous observation that sTMS affects a very small volume of cortex explains why a control site was not necessary in the current study: we assessed neuronal excitability by means of sTMS, and therefore moving the TMS coil to a different location would have made this assessment impossible. In theory, the possibility exists that the reduced neuronal excitability we measured under the TMS coil was an indirect effect caused by inactivation of an input area of PFG (e.g., neighboring area AIP or 7a). Even the inclusion of a remote control site would not entirely rule out this possibility because this control site would most likely not be connected to PFG and therefore would not cause any effect in PFG. We believe this theoretical possibility is unlikely because the induced electric field was maximal immediately under the coil where we recorded neuronal activity, and therefore this mechanism would imply the recruitment of another cortical area with a lower electric field. Nevertheless, cTBS and other reversible inactivation methods certainly evoke effects on connected remote areas (***Davare et al., 2010***; ***Bestmann et al., 2015***), sometimes even far away from the inactivation site (see, fe.g., ***Van Dromme et al., 2016*** for effects in inferior temporal cortex after reversible inactivation of AIP).

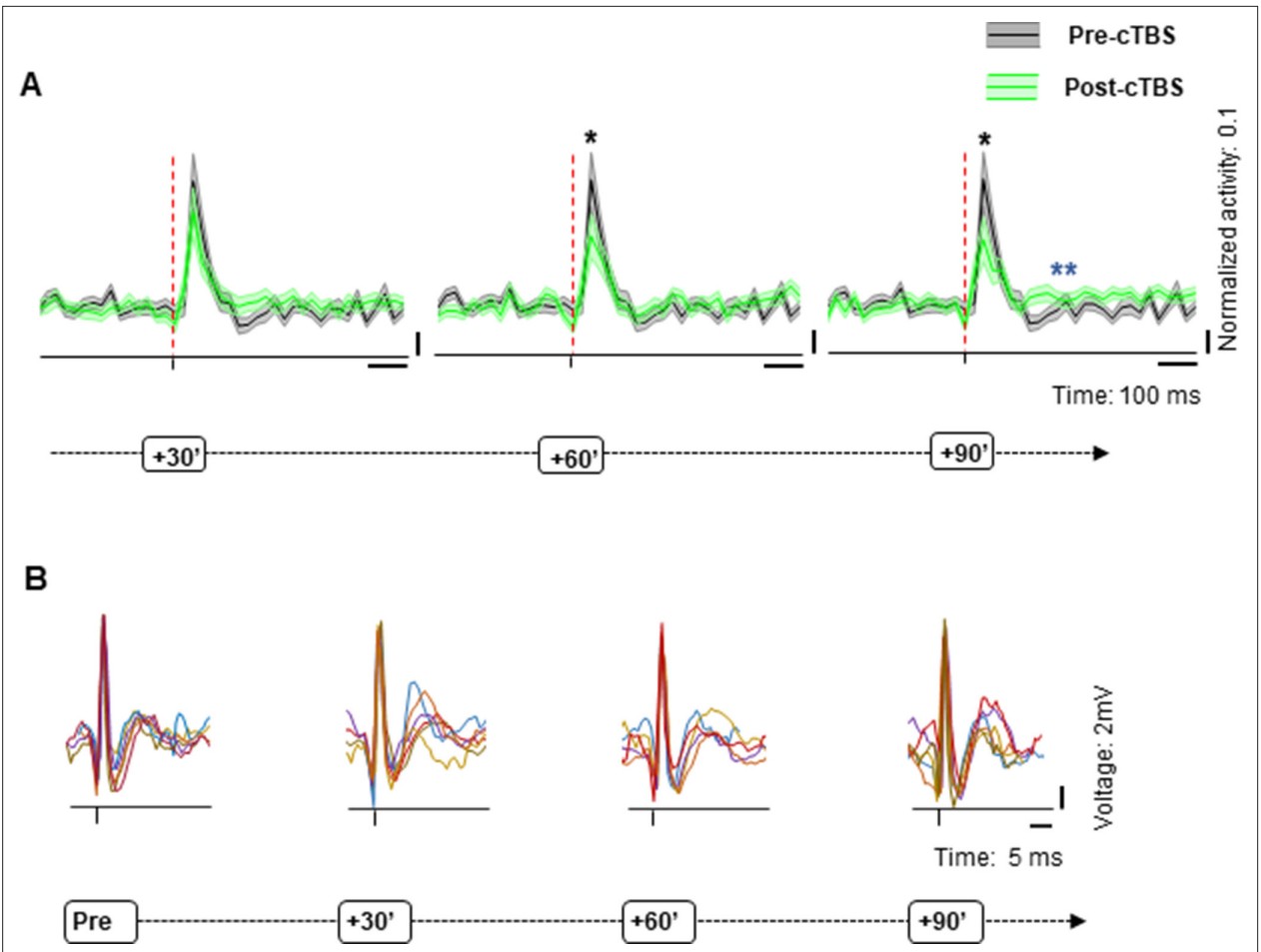

**Figure 8.** Population response to cTBS up to 90 min post-cTBS. (**A**) Average normalized (to the peak of the sTMS-evoked response) spike rate of all neurons recorded for up to 90 min post-cTBS (green) compared to pre-cTBS (black). Shading displays ±the standard error across neurons (N=34). Same conventions as in *Figure 7*. (**B**) Raw signal. Example voltage graph showing the pre- and post-cTBS spike waveforms from a representative neuron, monitored over time. In each panel, we plotted six consecutive spike waveforms (represented in different colors), recorded at four different time intervals (from left to right: pre-cTBS and 30, 60, and 90 min post-cTBS). cTBS, continuous theta-burst stimulation; sTMS, single-pulse transcranial magnetic stimulation.

It is important to note that we can rule out nonspecific factors influencing the neurons under the TMS coil. For example, cutaneous stimulation could not drive our effects since the TMS coil was positioned on the implant of the animal which was composed of dental acrylic. Moreover, we could measure the reduction in neuronal excitability 30–60 min after the administration of cTBS in task-related neurons in the absence of sTMS. It is highly unlikely that these late effects of cTBS would have resulted from cutaneous stimulation or other nonspecific factors. Note also that we previously observed highly grasp-specific effects of cTBS in line with the role of PFG in processing object properties for grasping (*Merken et al., 2021*).

Our results were robust (around 30% reduction in response) and highly reproducible in two animals, which is the standard in monkey electrophysiology experiments. Moreover, *Merken et al., 2021* also reported highly similar behavioral results in two different animals. However, future studies will have to determine to what extent the effects of cTBS are variable in a larger number of monkeys. Although the neural effects of cTBS were highly similar in the two animals we used, the effects of cTBS in human volunteers are notoriously variable across subjects (*Hamada et al., 2013*; *Hordacre et al., 2017*; *Jannati et al., 2017*). The reproducibility of our results was most likely related to the very controlled conditions in which we applied cTBS. In our setup, the TMS coil was rigidly anchored to the head implant of the animal, so that we kept both the position and the orientation of the coil similar across sessions. Also, another possibility is that monkeys become highly overtrained in the grasping task,

which may partially explain the similar behavioral effects of cTBS we reported in *Merken et al., 2021*. It is therefore plausible to assume that the larger variability inherent to human behavior is one reason underlying the variability of cTBS effects in humans, since stimulation is applied over a brain area in subjects at different levels of learning stages and behavioral performance, ultimately impacting on the susceptibility of that brain area to cTBS and increasing the inter-individual variability of the technique.

The critical advantage of the macaque monkey model is that we can measure single-cell activity and behavioral performance before and after cTBS. In a subset of neurons that we could record for more than 1 hr, we did not observe any recovery of the excitability even after 2 hr. Consistent with this observation, the behavioral effect of cTBS also lasted at least 2 hr (*Merken et al., 2021*). Therefore, our cTBS paradigm in monkeys may have caused longer-lasting effects than in human volunteers, possibly due to the thinner skull of the monkey, which could induce a stronger effect in parietal cortex. We did not observe any effect on the neuronal excitability on consecutive days, since we could always record single-unit activity over several weeks of recordings.

Investigating the neural effects of non-invasive brain stimulation techniques requires adequate animal models, so that behavioral measurements and detailed recordings of individual neurons can be combined with neuromodulation. In future studies, other stimulation protocols such as intermittent theta-burst stimulation and other neuromodulation techniques such as transcranial alternating current stimulation can be investigated and optimized in animal models using a similar approach.

## Materials and methods
### Subjects and surgical procedures
Two adult male rhesus monkeys (*Macaca mulatta*; monkey Y, 12 kg; monkey A, 8 kg) were trained to sit in a primate chair. A head post (Crist Instruments) was then implanted on the skull with ceramic screws and dental acrylic. For this and all other surgical protocols, monkeys were kept under propofol anesthesia (10 mg/kg/hr) and strict aseptic conditions. All experimental procedures were performed in accordance with the NIH's Guide for the Care and Use of Laboratory Animals and the EU Directive 2010/63/EU, and approved by the Ethical Committee at KU Leuven (project number P220/2014). Intensive training in passive fixation and visually guided grasping (VGG) began after 6 weeks of recovery. Once the monkeys had achieved an adequate level of performance, a craniotomy was made, guided by anatomical magnetic resonance imaging (MRI), over area PFG of the right hemisphere. An exhaustive description of this protocol has been detailed elsewhere (*Romero et al., 2019*). The recording chamber was implanted at a 45° angle with respect to the vertical, allowing oblique penetrations into the parietal convexity (*Figure 1A*). To confirm the recording positions, glass capillaries were filled with a 2% copper sulfate solution and inserted into a recording grid at five different locations during structural MRI (slice thickness: 0.6 mm). Two guiding rods were precisely fixed to the skull with dental acrylic based on the calculated stereotactic coordinates, allowing a highly reproducible positioning of the TMS coil across sessions. With these rods in place, the coil was kept at an angle of approximately 90° with respect to the recording chamber, inducing a posterior-anterior (PA) current over PFG. To estimate the center of stimulation, we used anatomical MRI and computed tomography (CT scan) to build 3D printed models of the skull and implant (*Figure 1B*). Based on the MR-CT co-registered images, we calculated that the TMS coil was placed approximately at a distance of 15 mm from (above) the parietal convexity.

### Stimulation protocol
For this study, we combined two different TMS protocols: sTMS and cTBS. We performed extracellular recordings before and after cTBS to investigate the changes in neuronal excitability, assessed with sTMS during passive fixation.

### cTBS effect on individual neurons: electrophysiological recordings
To optimize the stability of the recordings (total duration 1–3 hr per neuron), the animals performed a passive fixation task while sitting upright with their head fixed. With this setup, a single object (large sphere; diameter: 35 mm) was placed on a frontal plate, located 30 cm away from the animal. In the beginning of the trial, the monkeys remained in total darkness for a variable time period (intertrial interval: 2000–3000). Next, a red laser was projected at the base of the object, indicating the fixation

point. If the animal maintained fixation inside an electronically-defined window (±2.5° around the fixation point) for 500 ms, the object was illuminated (light onset) until the end of the trial (1300 ms). Whenever the monkey maintained a stable eye fixation, it received a drop of juice as reward.

Prior to the study, we determined the resting motor threshold (rMT) for each animal in a single session as the lowest stimulus intensity at which TMS applied over primary motor cortex (M1) produced a slight contralateral finger twitch 5 times out of 10 while the monkey held its hand in the resting position. For this measurement, the TMS coil (55 mm external diameter figure-of-eight, Magstim D25 branding iron style coil, with 25 mm windings ) was handheld over M1, at a distance of approximately 15 mm from the surface of the brain, reproducing the coil distance adopted during the experiments. The rMT was identical in the two animals (59% of the maximum stimulator output).

In each recording session, the coil was placed over the guiding rods, tangential to the skull and in contact with the implant, staying firmly anchored to the chair by means of an adjustable metal arm. We applied sTMS at 120% of the rMT (*Figure 1C*), aligned to light onset and randomly interleaved with no-stimulation trials, while advancing the electrode and searching for well-isolated units (Phase 1). We used a Magstim Rapid Stimulator (first generation stimulator, Magstim, UK), which applied biphasic pulses (100 µs rise time, 250 µs duration; 120% rMT corresponded to 70% of maximum stimulator output, for both monkeys) by means of a 55 mm figure-of-eight coil (Magstim D25 branding iron style coil). Note that the maximum stimulator output of first generation Magstim rapid stimulators is weaker and corresponds to 65% of that of a standard monophasic single-pulse Magstim stimulator (Magstim communication). As for previous experiments in our group, we employed this protocol to localize the center of stimulation, which was determined as the region under the center of the coil showing a significant sTMS-evoked effect in single neurons. With our technique, we isolated individual PFG neurons and stabilized the recordings to guarantee a reliable long-lasting monitoring of their activity. In each recording session, we first waited for a variable amount of time (40–60 min) to acquire the highest possible level of stability in the recorded signal and then measured the baseline neuronal activity and the sTMS-evoked response for 20 min. In order to prevent a possible overheating of the coil during this period, a gel-filled cool pack was placed around the handle, embracing the figure-of-eight-shaped head of the coil (Nexcare; 3M Company, MN). Next (Phase 2), we applied cTBS (*Figure 1C*) following the protocol described by *Huang et al., 2005*. In our cTBS paradigm, 50 Hz triplets were delivered at 200 ms intervals (300 pulses in total) for 20 s at 80% of the rMT (which corresponded to 47% of the maximum stimulator output intensity, for both monkeys). It is noteworthy to consider that our measured rMT intensity is likely to be underestimated, due to the monkeys not being entirely at rest during motor threshold definition procedures. Therefore, we assumed that our 80% rMT intensity should be comparable to the 70% rMT intensity used for cTBS in humans (*Goldsworthy et al., 2012*).

To prevent coil overheating during this phase, the gel-filled cool pack was now substituted by two round cool bags containing dry ice and placed over the loop ends of the coil. Phase three started immediately after cTBS, and consisted of a second period of combined sTMS and electrophysiological recordings, at 120% rMT for up to 2 hr post-cTBS to measure changes in neuronal excitability. For the entire duration of the experiments, none of the animals showed any noticeable side-effect to cTBS, performing their task without signs of distress.

We verified that the stimulation intensity we used was appropriate (i.e., sufficient to evoke behavioral effects but not too high) in two other rhesus monkeys in a VGG task (*Merken et al., 2021*). We measured significant increases in grasping time in both animals (53 and 41 ms on average in the interval between 20 and 120 min after cTBS), which were comparable in magnitude (approximately 15% increase in grasping time) to the effects of highly localized reversible inactivation experiments using muscimol in ventral premotor cortex (*Caprara and Janssen, 2021*). For comparison, reversible inactivation of primary motor cortex prolongs the grasping time by 43%. Therefore, the stimulation intensity we used for cTBS was sufficient to induce behavioral effects but did not induce a severe impairment in grasping.

We recorded single-unit activity in PFG using tungsten microelectrodes (impedance: 1 MΩ at 1 kHz; FHC, USA) inserted through the dura by means of a 23-gauge stainless steel guide tube and a hydraulic microdrive (FHC). Following the artifact reduction strategy proposed by *Mueller et al., 2014*, we used diodes and serial low-gain amplification to clip the artifact generated by the magnetic pulses, which prevented amplifier saturation. For this, we modified a regular BAK Electronics preamplifier (Model A-1; BAK Electronics, USA) by connecting two leakage diodes (BAS45A) anti-parallel

between the signal lines and ground before each stage of amplification. The initial front-end of the headstage remained unmodified to maintain the high-input impedance. With these settings, the duration of the evoked TMS artifact in our signal ranged from 8 to 12 ms (*Romero et al., 2019*). Neural activity was amplified and filtered (300–5000 Hz) following a standard recording protocol for spike detection. Using a dual time-window discriminator (LabVIEW and custom-built software), we isolated individual neurons and the TMS artifact, which was detected online and subtracted from the neural data. In addition, we recorded the entire raw signal (after filtering) for further analyses. Finally, we monitored the right eye position using an infrared-based camera system (Eye Link II, SR Research, Canada) sampling the pupil position at 500 Hz.

## Data analysis

All data analyses were performed in MATLAB (MathWorks, MA; code availability: DRYAD database). For the high- and low-stimulation trials of experiment 1, the neural activity was aligned on the sTMS pulse delivered at light onset. Also, for comparison, the no-stimulation trials were aligned on the same time bin. Net neural responses were then calculated as the average firing rate recorded after sTMS minus the baseline (spike rate calculated from the mean activity of the cell in the 800 ms interval preceding TMS).

We created line plots comparing the average response (spikes/s) of every cell during no-stimulation and stimulation (sTMS) trials recorded in the pre-cTBS epoch. The same analyses were then repeated in 10 min epochs post-cTBS, in order to assess in detail how the response of PFG neurons to sTMS—our measure of neuronal excitability—changed after cTBS. Since we expected that cTBS would induce cortical inhibition, we searched for PFG neurons showing excitatory responses to sTMS.

To determine the significance of the sTMS-evoked effect on individual neurons, we compared the cell responses observed in the first 40 ms after light onset in the stimulation condition to those in the no-stimulation condition (two-sided Wilcoxon rank sum test). To identify neurons with task-related activity, we ran a Wilcoxon signed rank test comparing the pre- and post-light onset responses in two different intervals (early task activity: 0–80 ms post-stimulus; later task activity: 100–500 ms post-stimulus) in the no-stimulation condition. Finally, to assess the effect of cTBS on neuronal excitability, we compared the average net sTMS-evoked response in every 10 min epoch after cTBS with the same response pre-cTBS (baseline sTMS net response, two-sided Wilcoxon signed rank test). For all neurons recorded for more than 1 hr, we compared the raw traces to verify (with visual inspection of the waveforms) that the neuron was not lost during the recordings.

## Acknowledgements

This work was supported by Fonds voor Wetenschappelijk Onderzoek Vlaanderen (Odysseus grants G.0007.12 and G.0C51.13N), Program Financing (PFV10/008) and KU Leuven grant C14/18/100. The authors would like to thank Stijn Verstraeten, Christophe Ulens, Piet Kayenbergh, Gerrit Meulemans, Marc De Paep, Astrid Hermans, and Inez Puttemans for their technical contributions.

## Additional information

### Funding

| Funder | Grant reference number | Author |
| --- | --- | --- |
| Fonds Wetenschappelijk Onderzoek | G.0007.12 | Peter Janssen |
| Fonds Wetenschappelijk Onderzoek | G.0C51.13N | Marco Davare |
| KU Leuven | PFV10/008 | Peter Janssen |
| KU Leuven | C14/18/100 | Peter Janssen |

The funders had no role in study design, data collection and interpretation, or the decision to submit the work for publication.

## Author contributions
Maria C Romero, Conceptualization, Data curation, Formal analysis, Investigation, Visualization, Methodology, Writing – original draft; Lara Merken, Data curation, Formal analysis, Investigation, Methodology; Peter Janssen, Conceptualization, Resources, Supervision, Funding acquisition, Writing – original draft, Project administration, Writing – review and editing; Marco Davare, Conceptualization, Resources, Writing – review and editing, Funding acquisition, Methodology

## Author ORCIDs
Maria C Romero ![ORCID] http://orcid.org/0000-0001-7758-2211
Peter Janssen ![ORCID] http://orcid.org/0000-0002-8463-5577
Marco Davare ![ORCID] http://orcid.org/0000-0002-4670-0251

## Ethics
This study was approved by the Committee on animal experiments of KU Leuven with number P220/2014.

## Decision letter and Author response
Decision letter https://doi.org/10.7554/eLife.65536.sa1
Author response https://doi.org/10.7554/eLife.65536.sa2

# Additional files

## Supplementary files
• Transparent reporting form

## Data availability
All data generated or analysed during this study are publicly available on Dryad https://doi.org/10.5061/dryad.ns1rn8pvr.

The following dataset was generated:

| Author(s) | Year | Dataset title | Dataset URL | Database and Identifier |
|---|---|---|---|---|
| Pita MC, Merken L, Janssen P, Davare M | 2022 | Neural effects of continuous theta-burst stimulation in macaque parietal neurons | https://doi.org/10.5061/dryad.ns1rn8pvr | Dryad Digital Repository, 10.5061/dryad.ns1rn8pvr |

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
