## [Editor Report]

This paper provides a fundamental advance on our understanding of the effects of a brain stimulation technique, continuous theta burst transcranial magnetic stimulation, that is widely used across a variety of subfields within human neuroscience. With convincing methodological rigor, the authors provide important validation of mechanism of action that produce the long-lasting effects of stimulation while simultaneously providing clues that speak to the variability observed in prior studies.

---

## [Decision Letter]

**Decision letter after peer review:**

Thank you for submitting your article "Neural effects of continuous theta-burst stimulation in macaque parietal neurons" for consideration by *eLife*. Your article has been reviewed by 2 peer reviewers, and the evaluation has been overseen by a Reviewing Editor and Chris Baker as the Senior Editor. The reviewers have opted to remain anonymous.

Essential revisions:

1) An issue identified by the reviewers was the lack of an adequate control condition for both experiments. Comparing no-stimulation to stimulation (Exp 1) and high vs low stimulus intensity (Exp 2) severely limits the conclusions that can be drawn from the data as presented. Absent an active control condition, it is unclear whether the results presented are specific to the site of stimulation and whether any behavioral effects of stimulation might be due to other non-specific effects of stimulation (e.g. somatosensory sensations).

2) All the reviewers agreed that the link between the single-unit recording experiment and the behavioral experiment was tenuous given they were recorded with different animals at different times with different protocols/conditions. Reviewers did not see the value of including the second experiment and suggested removing it.

3) There was substantial concern about the reporting and execution of the specific statistical analyses used. A re-analysis using methods that could be used to generalize to new animals/subjects is required as is a fuller reporting of the statistics (see reviewer comments below).

4) The exact stimulation intensities used and their relationship to the motor threshold of the different animals was unclear. There needs to be clarification of: the number of stimulation sessions, when motor thresholds were determined, the stimulation intensity used for each animal, etc.

*Reviewer #1 (Recommendations for the authors):*

This interesting work is a challenging study documenting at neuronal level the effect of rTMS in the excitability of neurons in the parietal cortex of the monkey. The study has been conducted in a rigorous way, with challenging techniques. All the data have been adequately shared and statistically treated. All conclusions are supported by the experimental data.

The paper sheds light on the foundations of a technique largely used in human volunteers to study neuronal circuits. They perform continuous theta burst stimulation over area PFG of the lateral parietal cortex to check the effect on neural excitability. This represents a technical advance in the field. The results represent the first demonstration of the effects of such widely used brain stimulation at individual neuronal level and will inform future clinical and basic research.

The study is performed in 4 monkeys, 2 studied in a fixation task and 2 in a grasping task. In the fixation task they show the time course of hypoexcitability, hyperexcitability and recovery time at individual neuronal level and at population level. A huge variability is demonstrated both in terms of time and of level of activation/depression.

In the grasping task, behavioural effects are shown, such as a long-lasting increase in the grasping time after TMS. The link between the 2 experiments is not straightforward, moreover the animals are not the same. Considering this and that the behavioural effects are part of Romero et al., 2019 and of another publication on the way, I strongly suggest to remove the grasping part from this manuscript. This would not reduce the large breath of this study. Indeed the neuronal effects shown in the first 2 animals are really foundational and complete. Alternatively, authors should better highlight the relationship between the two groups of results.

Another main suggestion is related to the exact site of TMS delivery. A very precise way to exactly reproduce the positioning of the coil has been set in place. This is really positive. I consider important to see the exact positioning of the coil. However in figure 1A, only a generic scheme is furnished. Please provide MRI with the reference for coil positioning, at least in this individual, better for both subjects.

Specific points to address:

1) Methods, how many recording sessions have been performed? was the rmt calculated before each experimental session? please clarify.

2) How the comparison between different waveform has been performed? By visual inspection or by applying some statistical methods?

3) Results, figure 5: how many neurons showed significant correlation coefficients in a given epoch?

It seems that the % of neurons with significant correlations increases over time. Please add data and a comment on this.

Then, authors should show the correlation coefficient in each plot in the figure, and propose a more interesting and convincing interpretation of the change of the correlation coefficient over time.

4) Results, figure 7. Authors should briefly comment the long-lasting duration of the response to object observation shown in figure 7.

5) Discussion, page 19. The various neural effects shown could be the explanation of the different and variable effects obtained in humans in theta-burst experiments. However, could it be that the different effects in humans could be related to the variability in kinematics across subjects, whereas the constant effect on monkey grasping time could be due to the stereotyped behavior of an overtrained monkey?

[Editors’ note: further revisions were suggested prior to acceptance, as described below.]

Thank you for resubmitting your work entitled "Neural effects of continuous theta-burst stimulation in macaque parietal neurons" for further consideration by *eLife*. Your revised article has been evaluated by Chris Baker (Senior Editor) and a Reviewing Editor.

The manuscript has been improved but there are some remaining issues that need to be addressed, as outlined below:

1) The lack of a control condition has not been adequately dealt with. At present it is not possible to determine whether the effects were specifically due to cTBS over parietal cortex. An active control condition with stimulation of another site would provide a comparison for how the cTBS protocol in general affects neuronal responses to spTMS. For example, how can the authors rule out that cutaneous stimulation (rather than the specific stimulation of neurons under the coil) is driving the changes in excitability? This is a major limitation of the study and should be clearly discussed and acknowledged.

2) Please provide more clarity on the stimulation intensity used. It is reported that both monkeys were stimulated at 47% MSO corresponding with 80% RMT. Was RMT calculated in each animal separately and they had the exact same RMT? Was an average RMT between the two animals used? Was there any measurement of the level of movement for each animal during thresholding? It would be helpful to be able to draw a clearer link between the stimulation intensity used here and the standard intensity used in humans.

3) Some of the statistical concerns raised by Reviewer #2 remain issues that need to be addressed. For each statistical test, all parameters (e.g. test statistic, degrees of freedom, effect size, etc) need to be reported rather than just the p-values. Please only report corrected statistics. There is still concern regarding the ANOVAs and degrees of freedom reported. It is not appropriate to pool measurements across both animals as if they were a single population of neurons nor should statistical tests be conducted across trials. This will require a reanalysis appropriate to the data collected.

[Editors’ note: further revisions were suggested prior to acceptance, as described below.]

Thank you for resubmitting your work entitled "Neural effects of continuous theta-burst stimulation in macaque parietal neurons" for further consideration by *eLife*. Your revised article has been evaluated by Chris Baker (Senior Editor) and a Reviewing Editor.

The manuscript has been improved but there are some remaining issues that need to be addressed, as outlined below:

1) Please explicitly acknowledge and discuss the problem of the specificity of the TMS effects. Although the current revision points to previous work describing the focality of single pulse TMS, the reviewers and I are still concerned with the specificity of the cTBS effect. Even if cutaneous stimulation could be ruled out, it cannot be determined from the present data whether the results of cTBS to that particular site or if similar results would have been obtained if cTBS is applied anywhere else in the brain. Regional specificity of the cTBS effect cannot be claimed as there is no comparison with a cTBS effect at a distal site.

2) Please explicitly acknowledge and discuss the limitations of the statistical analysis (see reviewer comments below).

3) Please clarify the characteristics of the TMS coil used (see reviewer comments below).

*Reviewer #2 (Recommendations for the authors):*

1) The problem of specificity has not been answered, aside from referring to another publication. The problem, for example, of skin stimulation is poorly dealt with. Figure 1A indicates clearly that the distance of the cortex from the coil is the same as the distance of the coil from the nearest cutaneous flap, so the presence of the resin support does not guarantee absence of cutaneous stimulation.

2) Regarding the problem of statistical analysis, the authors did not reply fully to the original question. I acknowledge that in neurophysiology it is customary to treat single neurons as subjects, but this procedure should be carefully explained. I understand the difficulty of training non-human primates and the invaluable insight of single neuron recordings, but the authors should also acknowledge the limitations of the interpretation of the data. I repeat myself in saying that the authors adopted a fixed effect analysis at group-level. Subjects (monkeys) are not treated as a random factor, so the level of inference is at the level of the group and cannot be extended to the population. An ANOVA with almost 5000 degrees of freedom and a very weak effect size (eta squared ranges from 0=no effect to 1=strong effect is reported to be = 0.01) should be reported cautiously in a generalist journal.

I propose to explicitly ask the authors to point out that the current statistics do not allow inference to the population.

In addition, I have a further comment on how the methods are reported:

3) The description of the coil characteristics is incorrect, and this ambiguity led to several misinterpretations. The coil is described in the methods as a 55 mm figure of eight coil. However, it is a universal convention in TMS to describe figure of eight coils by the outer diameter of each of the windings. Here it was used to describe the width of the whole coil. One realizes this only after taking the time to check the manufacturer's specifications. To confirm this, the legend of figure 1 reports the correct description, as a double 25 mm coil. This imprecision led to several misunderstandings (obviously a double 55 mm coil is too big for focality in a monkey brain, as I pointed out in my first review). I would suggest the authors be consistent in describing their methods and stick to international conventions. The problem of focality still persists, the intensity of stimulation indicated here are still very high, even with a small double 25 mm coil.

---

## [Author Response]

Essential revisions:1) An issue identified by the reviewers was the lack of an adequate control condition for both experiments. Comparing no-stimulation to stimulation (Exp 1) and high vs low stimulus intensity (Exp 2) severely limits the conclusions that can be drawn from the data as presented. Absent an active control condition, it is unclear whether the results presented are specific to the site of stimulation and whether any behavioral effects of stimulation might be due to other non-specific effects of stimulation (e.g. somatosensory sensations).

In addition to our detailed response to the reviewers, we would like to highlight that our ‘active stimulation’ condition in experiment 1 can be considered as including its own control: (1) our cTBS effect was unlikely to be an order (post- vs. pre-cTBS) effect since neuronal activity recovered to baseline (pre-cTBS) levels an hour post-cTBS. (2) cTBS did not always lead to an inhibition of neuronal firing rate, which argues against unspecific cTBS effects and (3) the firing rate inhibition post-cTBS occurred at different time points in different neurons, again arguing against a systematic non-specific effect of the stimulation.

In the present experiments, we did not investigate network (remote) effects of cTBS as this was beyond the scope of this study.

We have now added a paragraph in the Discussion to clarify these aspects to the reader.

2) All the reviewers agreed that the link between the single-unit recording experiment and the behavioral experiment was tenuous given they were recorded with different animals at different times with different protocols/conditions. Reviewers did not see the value of including the second experiment and suggested removing it.

Thank you for this suggestion, this experiment has now been removed. We only kept references to this behavioural work, which has now been published in more detail elsewhere.

3) There was substantial concern about the reporting and execution of the specific statistical analyses used. A re-analysis using methods that could be used to generalize to new animals/subjects is required as is a fuller reporting of the statistics (see reviewer comments below).

This has now been taken into account: the new analyses do not change our results and we compare both methods in a new table.

4) The exact stimulation intensities used and their relationship to the motor threshold of the different animals was unclear. There needs to be clarification of: the number of stimulation sessions, when motor thresholds were determined, the stimulation intensity used for each animal, etc.

Details of stimulation intensities and how they compare to human experiments have now been added.

Reviewer #1 (Recommendations for the authors):This interesting work is a challenging study documenting at neuronal level the effect of rTMS in the excitability of neurons in the parietal cortex of the monkey. The study has been conducted in a rigorous way, with challenging techniques. All the data have been adequately shared and statistically treated. All conclusions are supported by the experimental data.The paper sheds light on the foundations of a technique largely used in human volunteers to study neuronal circuits. They perform continuous theta burst stimulation over area PFG of the lateral parietal cortex to check the effect on neural excitability. This represents a technical advance in the field. The results represent the first demonstration of the effects of such widely used brain stimulation at individual neuronal level and will inform future clinical and basic research.The study is performed in 4 monkeys, 2 studied in a fixation task and 2 in a grasping task. In the fixation task they show the time course of hypoexcitability, hyperexcitability and recovery time at individual neuronal level and at population level. A huge variability is demonstrated both in terms of time and of level of activation/depression.In the grasping task, behavioural effects are shown, such as a long-lasting increase in the grasping time after TMS. The link between the 2 experiments is not straightforward, moreover the animals are not the same. Considering this and that the behavioural effects are part of Romero et al., 2019 and of another publication on the way, I strongly suggest to remove the grasping part from this manuscript. This would not reduce the large breath of this study. Indeed the neuronal effects shown in the first 2 animals are really foundational and complete. Alternatively, authors should better highlight the relationship between the two groups of results.

Thank you for this suggestion. We have now removed the behavioral experiment. Our main motivation was to show that the intensity of stimulation was not exaggerated for monkeys (see also comment 1 of reviewer 2). Therefore, we now merely mention the behavioral effects in the Methods on page 6 without discussing it further.

‘We verified that the stimulation intensity we used was appropriate (i.e. sufficient to evoke behavioral effects but not too high) in two other rhesus monkeys in a visually-guided grasping task. We measured significant increases in grasping time in both animals (53 ms and 41 ms on average in the interval between 20 and 120 min after cTBS), which were comparable in magnitude (approximately 15% increase in grasping time) to the effects of highly localized reversible inactivation experiments using muscimol in ventral premotor cortex (Caprara and Janssen, 2021). For comparison, reversible inactivation of primary motor cortex prolongs the grasping time by 43%. Therefore, the stimulation intensity we used for cTBS was sufficient to induce behavioral effects but did not induce a severe impairment in grasping.’Another main suggestion is related to the exact site of TMS delivery. A very precise way to exactly reproduce the positioning of the coil has been set in place. This is really positive. I consider important to see the exact positioning of the coil. However in figure 1A, only a generic scheme is furnished. Please provide MRI with the reference for coil positioning, at least in this individual, better for both subjects.

We have added an MRI with the dummy TMS coil in place in Figure 1A.

Specific points to address:1) Methods, how many recording sessions have been performed? was the rmt calculated before each experimental session? please clarify.

We have added the number of recording sessions (page 10) and specified that we determined the RMT once before all following recording sessions (page 5).

2) How the comparison between different waveform has been performed? By visual inspection or by applying some statistical methods?

We have now clarified that we visually inspected the spike waveforms.

3) Results, figure 5: how many neurons showed significant correlation coefficients in a given epoch?It seems that the % of neurons with significant correlations increases over time. Please add data and a comment on this.

Figure 5 shows the correlation between the reduction in baseline activity and the reduction in the sTMS evoked response across the population of neurons. Therefore, we cannot calculate the correlation for individual neurons. We observed a significant increase in the correlation (based on the confidence intervals), which we have now added in the text on p. 14-15.

Then, authors should show the correlation coefficient in each plot in the figure, and propose a more interesting and convincing interpretation of the change of the correlation coefficient over time.

We have added these correlation coefficients and the confidence intervals (in Table 2) and discussed them on page 14. We have added the correlation coefficient in each time epoch in Figure 5. The correlation may increase over time because the effect of cTBS becomes apparent in more neurons in either the baseline activity, the sTMS evoked response, or in both. In the last epoch (60 min postcTBS), the correlation decreased again because neurons may have been at different stages of recovery. (Page 14-15)

4) Results, figure 7. Authors should briefly comment the long-lasting duration of the response to object observation shown in figure 7.

We have added a comment on this response on page 19.

5) Discussion, page 19. The various neural effects shown could be the explanation of the different and variable effects obtained in humans in theta-burst experiments. However, could it be that the different effects in humans could be related to the variability in kinematics across subjects, whereas the constant effect on monkey grasping time could be due to the stereotyped behavior of an overtrained monkey?

This is indeed a very good suggestion. However, since we have now removed the behavioral experiment, we now refer to the Merken et al., study to introduce this possibility in the discussion on page 21.

‘However, another possibility is that monkeys become highly overtrained in the grasping task, which may partially explain the similar behavioral effects of cTBS reported in Merken et al., (2021).’

[Editors' note: further revisions were suggested prior to acceptance, as described below.]

The manuscript has been improved but there are some remaining issues that need to be addressed, as outlined below:1) The lack of a control condition has not been adequately dealt with. At present it is not possible to determine whether the effects were specifically due to cTBS over parietal cortex. An active control condition with stimulation of another site would provide a comparison for how the cTBS protocol in general affects neuronal responses to spTMS. For example, how can the authors rule out that cutaneous stimulation (rather than the specific stimulation of neurons under the coil) is driving the changes in excitability? This is a major limitation of the study and should be clearly discussed and acknowledged.

We have added a paragraph on page 16 in the discussion on this issue. We acknowledge that we did not control for remote or network effects of cTBS (p.13). However, we know from our previous study (Romero et al., 2019, Nat Comm) that single pulse TMS (sTMS) effects are highly focal. Since, in the present study, we assessed the effect of cTBS with sTMS, we would not have been able to assess the changes in neuronal excitability if we had moved the TMS coil to another location. This would have resulted in simply replicating our Nat Comm study. It is also important to mention that our study is not a behavioral study (where control sites are necessary to determine the specificity of the effect), and that we recorded from neurons located immediately under the TMS coil, where we know the effects of TMS are the strongest (see Romero et al., 2019 Nat Comm). Obviously, the cTBS we applied over parietal cortex may also have induced effects in remote areas (anatomically connected to the stimulated area), but this will not have influenced our recordings. We also clearly state that cutaneous stimulation could not be a factor in our experiments because the TMS coil was positioned on the implant of the animal (which is composed of dental acrylic). Therefore, we are entirely confident that the neural effects we measured immediately under the TMS coil were indeed specifically due to cTBS over parietal cortex. Finally, it is important to note that we showed clear dose-response effects in our previous behavioral paper (Merken et al., 2021, Scientific Reports) with different behavioral effects for no-stimulation, low-stimulation and high-stimulation, further showing the specificity of our cTBS effects.

p.16: ‘Furthermore, our previous observation that single-pulse TMS affects a very small volume of cortex explains why a control site was not necessary in the current study: we assessed neuronal excitability by means of single-pulse TMS, and therefore moving the TMS coil to a different location would have made this assessment impossible. It is important to note that we can rule out non-specific factors influencing the neurons under the TMS coil. For example, cutaneous stimulation could not drive our effects since the TMS coil was positioned on the implant of the animal which was composed of dental acrylic.’

2) Please provide more clarity on the stimulation intensity used. It is reported that both monkeys were stimulated at 47% MSO corresponding with 80% RMT. Was RMT calculated in each animal separately and they had the exact same RMT? Was an average RMT between the two animals used? Was there any measurement of the level of movement for each animal during thresholding? It would be helpful to be able to draw a clearer link between the stimulation intensity used here and the standard intensity used in humans.

We have clarified now that we assessed the rMT once in a single session in each animal, and that the rMTs were identical in the two animals (p. 4). References to standard protocols used in humans are also made.

3) Some of the statistical concerns raised by Reviewer #2 remain issues that need to be addressed. For each statistical test, all parameters (e.g. test statistic, degrees of freedom, effect size, etc) need to be reported rather than just the p-values. Please only report corrected statistics. There is still concern regarding the ANOVAs and degrees of freedom reported. It is not appropriate to pool measurements across both animals as if they were a single population of neurons nor should statistical tests be conducted across trials. This will require a reanalysis appropriate to the data collected.

We have added all statistical information requested (degrees of freedom, z-values, effect sizes with r values for Wilcoxon and η_p_^2^ for ANOVAs) throughout the manuscript. We have also added a sentence (on p. 11) explaining that we pooled all neurons because the results were highly similar (as we do in all other non-human primate studies), but that we also provide all statistics for the two animals separately (see also Figure 4 supplement 1 for data of the two animals separately).

In addition, we have re-analyzed and replotted the data averaged across neurons instead of across trials. This did not change the main results (see for example the effects beyond one hour on p. 13).

[Editors' note: further revisions were suggested prior to acceptance, as described below.]

Reviewer #2 (Recommendations for the authors):1) The problem of specificity has not been answered, aside from referring to another publication. The problem, for example, of skin stimulation is poorly dealt with. Figure 1A indicates clearly that the distance of the cortex from the coil is the same as the distance of the coil from the nearest cutaneous flap, so the presence of the resin support does not guarantee absence of cutaneous stimulation.

We have addressed this issue now extensively in the discussion on page 16-17, and we have provided argumentation why an indirect effect through another area seems unlikely. However, we also acknowledge that cTBS and other neuromodulation techniques evoke network effects.

‘In theory, the possibility exists that the reduced neuronal excitability we measured under the TMS coil was an indirect effect caused by inactivation of an input area of PFG (e.g., neighboring area AIP or 7a). Even the inclusion of a remote control site would not entirely rule out this possibility because this control site would most likely not be connected to PFG and therefore would not cause any effect in PFG. We believe this theoretical possibility is unlikely because the induced electric field was maximal immediately under the coil where we recorded neuronal activity, and therefore this mechanism would imply recruitment of another cortical area with a lower electric field. Nevertheless, cTBS and other reversible inactivation methods certainly evoke effects on connected remote areas (Davare et al., 2010; Bestman paper), sometimes even far away from the inactivation site (see for example Van Dromme et al. (2016) for effects in inferior temporal cortex after reversible inactivation of AIP).’

In addition, we have clarified the issue of potential nonspecific factors contributing to our effects on p 17. Even if we would have evoked cutaneous effects with cTBS, it is extremely unlikely that we would be able to observe a reduction in neuronal excitability 30 min later (as in the task-related neurons). For the reviewer, we also include Author response image 1, a picture of the setup to illustrate the positioning of the TMS coil.

‘It is important to note that we can rule out non-specific factors influencing the neurons under the TMS coil. For example, cutaneous stimulation could not drive our effects since the TMS coil was positioned on the implant of the animal which was composed of dental acrylic. Moreover, we could measure the reduction in neuronal excitability 30-60 min after the administration of cTBS in task-related neurons in the absence of sTMS. It is highly unlikely that these late effects of cTBS would have resulted from cutaneous stimulation or other nonspecific factors. Note also that we previously observed highly grasp-specific effects of cTBS in line with the role of PFG in processing object properties for grasping (Merken et al., 2021).’

**Author response image 1. sa2fig1:** 

2) Regarding the problem of statistical analysis, the authors did not reply fully to the original question. I acknowledge that in neurophysiology it is customary to treat single neurons as subjects, but this procedure should be carefully explained. I understand the difficulty of training non-human primates and the invaluable insight of single neuron recordings, but the authors should also acknowledge the limitations of the interpretation of the data. I repeat myself in saying that the authors adopted a fixed effect analysis at group-level. Subjects (monkeys) are not treated as a random factor, so the level of inference is at the level of the group and cannot be extended to the population. An ANOVA with almost 5000 degrees of freedom and a very weak effect size (eta squared ranges from 0=no effect to 1=strong effect is reported to be = 0.01) should be reported cautiously in a generalist journal.I propose to explicitly ask the authors to point out that the current statistics do not allow inference to the population.

We have addressed this issue (which is inherent to almost every NHP study in our domain) in the discussion (p.17). The ANOVA on the baseline effect is now replaced by a two separate Wilcoxon ranksum tests (for the two animals separately) on p. 11.

‘Our results were robust (around 30% reduction in response) and highly reproducible in two animals, which is the standard in monkey electrophysiology experiments. Moreover, Merken et al., (2020) also reported highly similar behavioral results in two different animals. However, future studies will have to determine to what extent the effects of cTBS are variable in a larger number of monkeys.’

p.11 ‘two-sided Wilcoxon ranksum tests comparing the pre-cTBS cell response with that measured at 40 min post-cTBS; for the two animals at 40 min: z = 2.66, p = 0.01, r = 0.26 for monkey Y and z = 1.86, p = 0.06, r = 0.22 for monkey A’.

In addition, I have a further comment on how the methods are reported:3) The description of the coil characteristics is incorrect, and this ambiguity led to several misinterpretations. The coil is described in the methods as a 55 mm figure of eight coil. However, it is a universal convention in TMS to describe figure of eight coils by the outer diameter of each of the windings. Here it was used to describe the width of the whole coil. One realizes this only after taking the time to check the manufacturer's specifications. To confirm this, the legend of figure 1 reports the correct description, as a double 25 mm coil. This imprecision led to several misunderstandings (obviously a double 55 mm coil is too big for focality in a monkey brain, as I pointed out in my first review). I would suggest the authors be consistent in describing their methods and stick to international conventions. The problem of focality still persists, the intensity of stimulation indicated here are still very high, even with a small double 25 mm coil.

We have clarified this now on page 4.

‘…the TMS coil (55 mm external diameter figure-of-eight, Magstim D25 branding iron style coil**,** with 25 mm windings )’.